# Learning Curve Analysis of Robotic-Assisted Mitral Valve Repair with COVID-19 Exogenous Factor: A Single Center Experience

**DOI:** 10.3390/medicina59091568

**Published:** 2023-08-29

**Authors:** Laura Giroletti, Valentina Brembilla, Ascanio Graniero, Giovanni Albano, Nicola Villari, Claudio Roscitano, Matteo Parrinello, Valentina Grazioli, Ettore Lanzarone, Alfonso Agnino

**Affiliations:** 1Division of Robotic and Minimally Invasive Cardiac Surgery, Humanitas Gavazzeni Hospital, 24125 Bergamo, Italy; ascanio.graniero@gavazzeni.it (A.G.); alfonso.agnino@gavazzeni.it (A.A.); 2Department of Management, Information and Production Engineering, University of Bergamo, 24044 Dalmine (Bg), Italy; v.brembilla1@studenti.unibg.it (V.B.); ettore.lanzarone@unibg.it (E.L.); 3Division of Cardiac Anesthesia, Humanitas Gavazzeni Hospital, 24125 Bergamo, Italy; giovanni.albano@gavazzeni.it (G.A.); nicola.villari@gavazzeni.it (N.V.); claudio.roscitano@gavazzeni.it (C.R.); matteo.parrinello@gavazzeni.it (M.P.); 4Cardiovascular Surgery Department, Humanitas Gavazzeni Hospital, 24125 Bergamo, Italy; valentina.grazioli@gavazzeni.it

**Keywords:** mitral valve repair, robotic-assisted mitral valve surgery (RAMVS), learning curve, COVID-19 pandemic, cardiac surgery team

## Abstract

*Background and objective* Renewed interest in robot-assisted cardiac procedures has been demonstrated by several studies. However, concerns have been raised about the need for a long and complex learning curve. In addition, the COVID-19 pandemic in 2020 might have affected the learning curve of these procedures. In this study, we investigated the impact of COVID-19 on the learning curve of robotic-assisted mitral valve surgery (RAMVS). The aim was to understand whether or not the benefits of RAMVS are compromised by its learning curve. *Materials and Methods* Between May 2019 and March 2023, 149 patients underwent RAMVS using the Da Vinci^®^ X Surgical System at the Humanitas Gavazzeni Hospital, Bergamo, Italy. The selection of patients enrolled in the study was not influenced by case complexity. Regression models were used to formalize the learning curves, where preoperative data along with date of surgery and presence of COVID-19 were treated as the input covariates, while intraoperative and postoperative data were analyzed as output variables. *Results* The age of patients was 59.1 ± 13.3 years, and 70.5% were male. In total, 38.2% of the patients were operated on during the COVID-19 pandemic. The statistical analysis showed the positive impact of the learning curve on the trend of postoperative parameters, progressively reducing times and other key indicators. Focusing on the COVID-19 pandemic, statistical analysis did not recognize an impact on postoperative outcomes, although it became clear that variables not directly related to the intervention, especially ICU hours, were strongly influenced by hospital logistics during COVID-19. *Conclusions* Understanding the learning curve of robotic surgical procedures is essential to ensure their effectiveness and benefits. The learning curve involves not only surgeons but also other health care providers, and establishing a stable team in the early stage, as in our case, is important to shorten the duration. In fact, an exogenous factor such as the COVID-19 pandemic did not affect the robotic program despite the fact that the pandemic occurred early in the program.

## 1. Introduction

The first robotic surgical procedure for cardiac surgery was performed in 1998, but despite initial enthusiasm in the early 2000s, the growth of this approach in Europe has not been very rapid. There were several reasons for this: immature technology, high cost, and a long learning curve compared to the traditional thoracotomy approach were the main critical issues. However, Pettinari et al. [1] showed a renewed interest since 2011 in the robotic-assisted procedure, especially in the field of mitral valve (MV) repair. More recently, in 2020, Cerny et al. [2] observed a 112% increase in annual robotic surgery volumes during their 4-year study, from 435 in 2016 to 923 in 2019. The improvement in robotic technology has played a key role in this change. Modern robotic tele-manipulators provide three-dimensional and magnified visualization (10×). In addition, they are equipped with articulated instruments with seven degrees of freedom of movement, thus significantly optimizing dexterity. In this context, MV repair is the perfect and ideal setting for the application of the robotic approach [3], the safety, efficacy and advantages of which have already been described in the literature [3,4,5,6,7].

A much-debated point is the high cost of this procedure. Several works have suggested that the benefits of this approach (including reduced blood transfusions, reduced length of stay in the ICU and hospital, less invasiveness, and reduced wound problems) may justify and mitigate its cost [8,9,10]. The learning curve is another major concern for robotic surgery, because of its complexity and duration, and because it involves not only cardiac surgeons but also an entire medical team. For this reason, most cardiac surgery centers are reluctant to adopt this approach. In addition, the COVID-19 pandemic in 2020 might have affected the learning curve, especially in centers that had recently started a robotic program.

There is a dearth of studies on this topic in the medical literature, and none of the studies have considered the role of the COVID-19 pandemic. In our work, we investigated the impact of COVID-19 and several other factors on the learning curve, i.e., on intraoperative and postoperative outcomes of patients undergoing robotic-assisted MV surgery (RAMVS). We refer in this study to the case of the Cardiac Surgery Center of the Humanitas Gavazzeni Hospital, Bergamo, Italy, which is located in one of the regions that was most affected by the COVID-19 pandemic not only in Italy but also in Europe, and which had started RAMVS procedures shortly before the onset of the pandemic.

## 2. Materials and Methods

### 2.1. Study Population

The data analyzed in this study were obtained from the database of the Humanitas Gavazzeni Hospital, Bergamo, Italy. Between May 2019 and March 2023, 149 patients underwent RAMVS using the Da Vinci^®^ X Surgical System (Intuitive Surgical Inc., Sunnyvale, CA, USA). In this study, we included all patients affected by MV dysfunction (regurgitation or stenosis) regardless of etiology, type of surgical technique (different types of MV repair or replacement), type of aortic cross-clamping (external or endoaortic), and type of mitral surgery with combined procedures, such as the closure of interatrial septal defects or of the left atrial appendage. Patients with coronary artery disease were not considered.

All patients underwent a preoperative diagnostic workup that consisted of computed tomography (CT) scan of the chest, abdomen and pelvis, coronary angiography, transthoracic or transesophageal echocardiography, electrocardiography, and routine biochemical tests. Moreover, they underwent CT angiography to better study the anatomy and have sufficient intrathoracic clearance for the navigation of the robotic arms. In the postoperative period, all patients underwent regular examinations according to the course: biochemical tests, chest radiography, echocardiography or CT scan.

### 2.2. Surgical Technique

The adopted RAMVS technique has been previously described [3]. Briefly, an incision of 1.5–2 cm was performed in the third intercostal space for access to the working port, and four other incisions of 0.8 cm were performed as instrumental ports (camera arm, right and left arm, and dynamic left arm). Cardiopulmonary bypass (CPB) was established peripherally via arterial and femoral cannulation. Cold crystalloid cardioplegia (HTK solution, Custodiol, Franz Köhler Chemie GmbH, Bensheim, Germany) was used for all patients. Cross aortic clamping was achieved with an external Chitwood device or an endoaortic balloon according to the patient’s anatomical characteristics. Different surgical techniques were employed for MV repair, including quadrangular or triangular resection of the posterior leaflet, sliding, edge to edge, neo-chordae implant, and the “Lavaredo technique” [11]. Annuloplasty with semi-rigid or flexible ring was performed in all patients. Mechanical or biological prostheses were used according to clinical parameters.

### 2.3. Collected Data

All 149 patients mentioned above were included in the study. The selection of patients operated on with RAMVS and, therefore, included in this study was not influenced by the complexity of the case, i.e., there was no selection of the simplest cases in the early stages of the robotic program. Therefore, there was no trend in the complexity of the cases over time, and we could argue that any negative or positive trends observed in the learning curves depended exclusively on the progressive accumulation of experience and dexterity in the procedure, or on exogenous factors such as the COVID-19 pandemic.

The data collected referred to the three phases of hospital stay: preoperative, intraoperative and postoperative. They are detailed below.

#### 2.3.1. Preoperative Data

These data cover the patient’s health status at the time of admission and in the days before surgery, the day the surgery was performed, and the COVID-19 situation on the surgery day. They include:*Male*: binary variable equal to 1 if male, and 0 if female;*Date* of the intervention, expressed as number of days since the first RAMVS surgery (performed on 9 May 2019 and with *Date* = 0);*COVID*: binary variable indicating the altered clinical activities due to the COVID-19 pandemic, which is equal to 1 from *Date* = 285 (19 February 2020) to *Date* = 957 (21 December 2021). It is worth noting that the first intervention with *COVID* = 1 was performed on *Date* = 397, showing an outage of activities of more than 100 days.*Age* in years;*Height* in centimeters;*Weight* in kilograms;*BSA* (body surface area), equal to 0.007814·*Height*^0.725^*·Weight*^0.425^;*BMI* (body mass index) equal to *Weight*/*Height*^2^;*Hb-pre* (preoperative hemoglobin) expressed in *mg/dL*;*Rhythm*: binary variable equal to 1 if the patient’s cardiac rhythm is not sinusal, and 0 if sinusal;*HTA*: binary variable equal to 1 if the patient has arterial hypertension, and 0 otherwise;*Diab-NID*: binary variable equal to 1 if the patient is a non-insulin dependent diabetic patient, and 0 otherwise;*Diab-ID*: binary variable equal to 1 if the patient is an insulin-dependent diabetic patient, and 0 otherwise;*Resp-ins*: binary variable equal to 1 if the patient has respiratory failure, and 0 otherwise;*Smoke*: binary variable equal to 1 if the patient is a smoker or an ex-smoker, and 0 otherwise;*Endoc*: binary variable equal to 1 if the patient has active endocarditis, and 0 otherwise;*ASA* (American Society of Anesthesiology score): categorical variable with 4 levels, from I to IV, which determines if the patient is healthy enough to tolerate surgery and anesthesia;*EuroscoreII* (European System for Cardiac Operative Risk Evaluation): numerical score based on 17 parameters indicating the risk of death from heart surgery [12];*NHYA* (New York Heart Association classification): categorical variable with 4 levels, from I to IV, which represents the heart failure intensity based on the activities the patient is able to perform;*EF-pre* (preoperative ejection fraction): ratio between stroke volume and end-diastolic volume for the left ventricle*Mitral-reg* (severity of mitral regurgitation): categorical variable with 3 levels, from I to III;*Mitral-st*: binary variable equal to 1 if the patient has mitral stenosis, and 0 otherwise;*Pulm-hypert*: binary variable equal to 1 if the patient has pulmonary hypertension, and 0 otherwise;*AoR*: binary variable equal to 1 if the patient has aortic regurgitation, and 0 otherwise;*Creatinine*, expressed in mg/dL;*Dialysis*: binary variable equal to 1 if the patient is currently on dialysis, and 0 otherwise.

In the learning curves, these variables were all treated as input covariates.

#### 2.3.2. Intraoperative Data

The following intraoperative data were recorded, taken from the operating room register:*CPB-time* (duration of CPB), expressed in minutes;*Clamp-time* (clamping time of the aorta), expressed in minutes;*StS-time* (skin-to-skin time elapsed from incision to suture), expressed in minutes;*TOR-time* (total operating time elapsed from patient entry into the operating room to exit), expressed in minutes;*OR-extub*: binary variable equal to 1 if the patient is extubated directly on the operating table without the need for intubation in the ICU, and 0 otherwise.

They represent performance data; therefore, they were all treated as output variables of a specific learning curve. They are all indicative of the team’s experience, although they also depend on the patient’s health status.

#### 2.3.3. Postoperative Data

The following postoperative data were considered in the study:*Bleeding-24h*: volume of fluid collected from the drains in the 24 post-operative hours, expressed in cc (not exclusively blood);*Transfus*: binary variable equal to 1 if the patient has been transfused, including during CPB, and 0 otherwise;*N-transf*: overall number of blood units received by the patient, including during CPB;*Hb-post*: postoperative hemoglobin the day after surgery, expressed in mg/dL;*ICU-hours*: time in hours spent by the patient in the ICU;*Post-days*: number of post-operative hospitalization days including the ICU;*Hosp-days*: total hospitalization days including preoperative stay;*Home*: binary variable equal to 1 if the patient directly returns home upon discharge, and 0 if he/she needs to stay at a rehabilitation center.

They represent the effectiveness of the surgery; therefore, similarly to the intraoperative data, they were all treated as output variables of a specific learning curve.

### 2.4. Statistical Analyses

Continuous variables are reported as mean ± standard deviation (SD) when they are normally distributed (Shapiro–Wilk test *p*-value greater than 0.05), and as median along with the 25th and 75th percentiles in square brackets otherwise. Binary variables are reported as percentages of positive cases. Categorical variables are reported as percentages of the observed levels.

According to the methodology used in the literature to model learning curves [13,14], each output variable was linked to the input covariates by means of a regression model. The type of regression was chosen according to the type of output variable: a logit regression was used for binary variables (*OR-extub*, *Transfus* and *Home*), a gamma regression was used for numerical positive variables (*CPB-time*, *Clamp-time*, *StS-time*, *TOR-time*, *Bleeding-24h*, *Hb-post* and *ICU-hours*), and a Poisson regression was used for count variables (*N-transf*, *Post-days*, and *Hosp-days*).

All covariates were included in a linear formulation without interaction terms. Moreover, the data were analyzed in order to reveal clear nonlinear trends. This analysis showed that *Date*, *Age*, *BSA*, *EuroscoreII* and *Creatinine* exhibited a nonlinear trend with respect to at least one of the output variables. Therefore, to give flexibility to the regression models, the associated squared values of these variables were added among the covariates. Hereafter, they are denoted as *Date^2^*, *Age^2^*, *BSA^2^*, *EuroscoreII^2^* and *Creatinine^2^*, respectively.

In addition, for each output variable, a model reduction was performed using the Akaike information criterion (AIC) [15]. Accordingly, only the subset of covariates and squared covariates resulting from the AIC was included in each learning curve presented in the results.

The quality of fitting was finally evaluated by means of McFadden’s pseudo-*R^2^* coefficient, for which a value above 0.4 could be interpreted as showing a high fit to the data [16].

Regressions were implemented in R via the function *glm*, using *binomial(link = ‘logit’)* as family for the binary variables, *gamma(link = ‘log’)* for the numerical positive variables, and *poisson(link = ‘log’)* for the count variables. The AIC was implemented via the function *stepAIC*.

## 3. Results

The number of missing values was 11 for the postoperative variables *Transfus* and *N-transf*, while there were no missing data for all other variables.

As expected, the number of surgeries increased over time after the pandemic: from June 2020 to March 2021, there were an average of 2–3 interventions per month, while later, the frequency increased up to 6–7 interventions per month.

A summary of the collected data is shown in Table 1. Of note, the quantiles of *N-transf* were all equal to zero because 104 observations out of 138 had *N-transf* = 0.

Table 2, Table 3 and Table 4 detail the regression models of each output variable when only the covariates defined by the AIC are included: Table 2 refers to the intraoperative variables, while Table 3 and Table 4 refer to the postoperative variables. The absence of a value in a cell indicates that the covariate was excluded by the AIC, and when a covariate was excluded from all models of a table, the row of that covariate was not even reported. Otherwise, for the included covariates, the coefficient of the regression model is reported along with the significance level. In the case of the categorical covariates, the coefficients for all levels other than the first are reported, since the categorical covariates with *K* levels were modeled as *K*-1 binary covariates describing whether the level was one between the second and the last.

In the case of the numerical covariates, a positive coefficient value, especially when associated with a high significance level, means that the output variable is positively correlated with the covariate, and vice versa. In other words, when the covariate increases, the output variable increases as well and vice versa. For the variables included with a quadratic value (*Date*, *Age*, *BSA*, *EuroscoreII* and *Creatinine*), the coefficients refer to the “shape” of the learning curve; specifically, the coefficient of the quadratic value refers to the concavity/convexity of the curve, while the coefficient of the value as such denotes the slope near the origin where the value of the covariate is 0. As for the binary covariates, a positive coefficient means that the covariate causes the output variable to increase and vice versa.

As expected, *Resp-ins*, *Endoc* and *Dialysis* were always excluded by the AIC because they were never observed in the study population (incidence of 0%).

The pseudo-*R^2^* values were not very high in the regressions, ranging from 0.127 to 0.569, which could be explained by the limited number of patients included and the inter-patient variability. Nevertheless, the AIC suggested the presence of covariates that are important in describing each output variable, whose coefficients are often significant in the models. Therefore, we can focus on these variables and draw conclusions at least regarding the direction of their impact.

For a visual representation of these results, Figure 1 shows the covariates that were included by the AIC and had a relevant significance level, i.e., a *p*-value less than 0.05. A green cell indicates a relevant favorable effect (i.e., a reduction in *CPB-time*, *Clamp-time*, *StS-time*, *TOR-time*, *Bleeding-24h*, *Transfus*, *N-transf*, *ICU-hours*, *Post-days* or *Hosp-days*, an increase in *OR-extub*, *Hb-post* or *Home*), while a red cell indicates a relevant unfavorable effect (i.e., vice versa). For the covariates included with a quadratic value, the overall effect in the neighbor of the observed values (see Table 1) is considered, while for the categorical covariates, the effect is reported in terms of belonging to higher classes with respect to smaller.

We first focused on the two main covariates of interest for our analysis (*Date* and *COVID*) and analyzed their impact. As for *Date* (numerical covariate with the associated squared covariate), small but significant coefficients were generally obtained. For some output variables, coefficients revealed a decreasing trend, with a steeper slope at the beginning, up to the vertex of the parabola on *Date* ≈ 950 (i.e., the median of the observations). This was the case for *CPB-time*, *Clamp-time*, *StS-time*, *TOR-time*, *OR-extub* and *ICU-hours*. The same result was found for *Transfus* and *Hosp-Days* but with a vertex on *Date* ≈ 580 and *Date* ≈ 790, respectively. In the case of *Hb-post* and *Post-days*, only a positive coefficient for the quadratic term was included by the AIC, highlighting a convex increasing trend. Finally, the AIC did not indicate any role of *Date* for *Bleeding-24h*, *N-transf* and *Home*. As for *COVID* (binary input covariate), the pandemic affected the learning curve of a few output variables. Indeed, increased values of *Hb-post* and decreased values of *Bleeding-24h* and *N-transf* were described by the regressions in the presence of *COVID* = 1.

It is worth remarking that the *ICU-hours* variable was strongly influenced by hospital logistics during the COVID-19 period, and not just decided based on the actual patient needs due to the postoperative course; in any case, *COVID* was not recognized by the AIC to play a role for this output variable.

Other preoperative covariates included in the regressions also showed trends that are worthy of discussion. *Age* was significant for *Post-days and Transf*, which generally increased with advancing age. Higher *NYHA* classes (*II* and *III*) reduced *Post-day and Hosp-day,* in line with clinical practice. *Weight* increased *StS-time* and *TOR-time* values, while *BSA* decreased them; these trends, which may seem conflicting, are discussed in more detail in the next section. *Diab-ID* significantly increased *TOR-time*. High *EuroscoreII* classes were associated with greater bleeding volumes (*Bleeding-24h*), resulting in more transfusions (*N-transf*).

Other covariates showed reduced significance.

## 4. Discussion

In the past decade, the growing interest in robotic-assisted cardiac surgery has drawn attention to its learning curve, especially in centers that started the procedure for the first time. As noted by other authors who analyzed the learning curve at their institution [17], in many hospitals, MV repair is still performed via conventional sternotomy. Although the repair technique is the same, the setting and challenges of these two strategies are very different. This means that an experienced surgeon approaching the robotic technique must often change his/her state of mind. In our study, the robotic program was initiated by a surgeon experienced in minimally invasive cardiac surgery, who was used to working in confined spaces. To perform safe and effective procedures, dedicated formal training with simulators (to obtain certifications for robotic surgery) and the assistance of experienced operators during the initial phase are also necessary [2,18]. We believe that this phase is very important and influences the duration and quality of the learning curve itself.

Moreover, training and learning curve affect not only surgeons, but also the entire medical team, including nurses, perfusionists and anesthesiologists. For this reason, it is important to create a stable and coordinated surgical team that does not change during the first phase of the program [19]. Like other authors [17], we believe that a stable team can affect outcomes and the learning curve itself, which is longer if members change frequently.

Our analyses showed that the preoperative covariate *Date* influenced some important surgical outcomes with a parabolic trend, such as *CPB-time*, *Clamp-time*, *StS-time*, *TOR-time*, *OR-extub* and *ICU-hours*. The obtained coefficients revealed a decreasing trend for these outcomes, with a steeper slope at the beginning. Similar to what has been observed in other studies [17,20], the greatest impact on the learning curve occurred in the early cases, and then a plateau was reached, which was dictated by the same “technical times” present in classical MV surgery. The same was found for *Transfus* and *Hosp-Days*. To further improve surgical outcomes after the plateau, we believe that technological advancement of the robotic device may be a relevant factor.

Eight months after the start of our robotic program, the COVID-19 pandemic radically changed the logistical organization of our hospital; therefore, we included this dramatic experience in the study through an additional input covariate. However, the analyses showed that the pandemic did not significantly affect the learning curve. We hypothesize that a good initial training and the creation of a stable and strong team, as already established in our case, might have played an important role in preventing the negative influence of this external factor and its consequences on the learning curve.

Regarding the other covariates, patient *Age* had a significant impact on postoperative clinical outcomes in terms of hospitalization and blood transfusions, which increased in elderly patients similarly to traditional cardiac surgery and common clinical practice. Also, *EuroscoreII*, which was correlated with *Age* in our study population, was associated with higher bleeding in the 24 hours after surgery (*Bleeding-24h*), resulting in more transfusions (*N-transf*).

Another comment is about diabetes, which significantly increased *TOR-time*; diabetes is known to be related to the development of arterial disease, and RAMVS is performed using peripheral arterial and femoral cannulation to establish CPB. We noted that, in diabetic patients, the poorer quality of femoral arteries requires longer isolation and cannulation times, as well as possible reconstruction of vessels due to calcified plaques or wall fragility, thereby lengthening the operation time. In addition, peripheral arterial disease can make it difficult to place the vascular accesses for anesthesia management, thus prolonging *TOR-time*.

Another interesting result was the role of *Weight* and *BSA* in postoperative *StS-time* and *TOR-time*: *Weight* increased their values, while *BSA* decreased them. This apparent contradiction can be explained by considering, for the same *Weight*, that a tall patient with a higher *BSA* has a slimmer phenotype and a better distribution of adipose tissue, which facilitates port positioning, the movement of robotic arms, and peripheral cannulation. Likewise, the greater the *Weight* in patients of short stature and low BSA, the greater the technical difficulty and timing of the surgery. All of these results supported our initial patient selection strategy based on chest anatomical characteristics and not on the type of valve disease. To initiate robotic surgery, we preferred normal weight patients with standard *BSA*, and we later extended the technique to patients with a small BSA and patients who were overweight.

### Limitations

The main limitation of this study is the small number of patients included. Therefore, the analyses can only be viewed as descriptive of the specific case. On the one hand, this prevented us from making predictions about other situations, while on the other hand, it allowed us to study the specific and relevant case of a cardiac surgery center that started the RAMVS program shortly before the COVID-19 pandemic and is located in the heart of one of the largest and most critical red zones in Europe.

Additional data would have allowed us to use more sophisticated approaches; however, the performed regressions are in line with the analyses conducted in the literature to model learning curves in medical practice.

## 5. Conclusions

The growing interest in robotic-assisted cardiac surgery in Europe has drawn attention to the importance of its learning curve. Investing in this area is essential to ensure the effectiveness of the robotic surgical procedure and its benefits, as shown in our study. The learning curve involves not only surgeons but also other health care professionals, and establishing a stable team in the early stage is important to shorten the durations and improve outcomes. At the same time, a good initial training is essential to avoid initial inefficiencies. Indeed, in our experience, an exogenous factor such as the COVID-19 pandemic did not affect our robotic program because it was set up correctly in these respects.

## Figures and Tables

**Figure 1 medicina-59-01568-f001:**
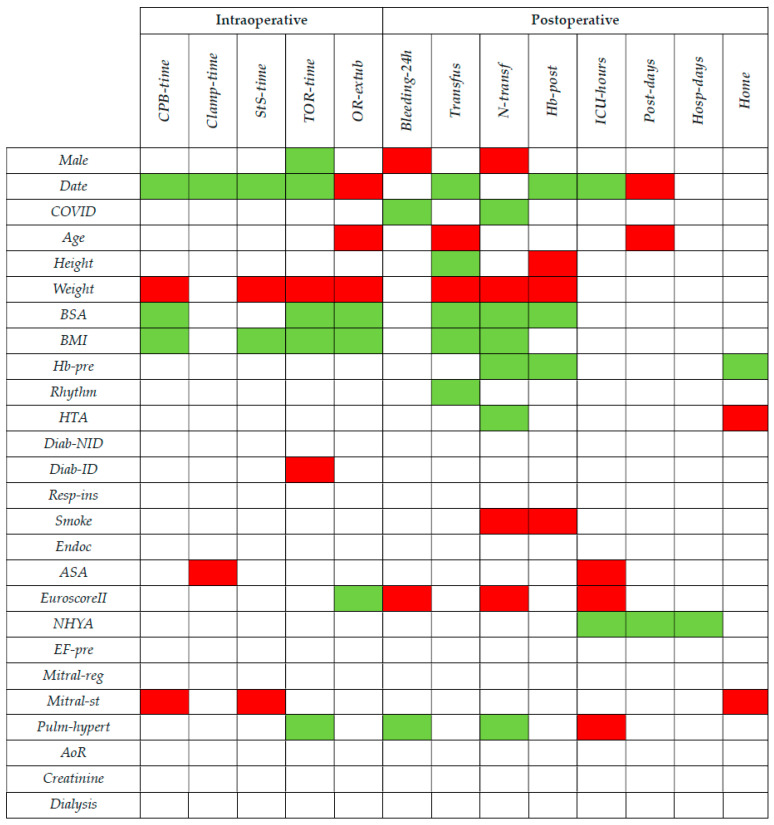
Covariates included by the AIC and with *p*-value less than 0.05. A green cell indicates a relevant favorable effect (i.e., a reduction in *CPB-time*, *Clamp-time*, *StS-time*, *TOR-time*, *Bleeding-24h*, *Transfus*, *N-transf*, *ICU-hours*, *Post-days* or *Hosp-days*, an increase in *OR-extub*, *Hb-post* or *Home*) while a red cell indicates a relevant unfavorable effect (i.e., vice versa).

**Table 1 medicina-59-01568-t001:** Summary of collected data, divided into preoperative, intraoperative and postoperative, reported as described in Section 2.4.

Preoperative Data			Intraoperative Data		Postoperative Data	
Variable		Value	Variable	Value	Variable	Value
*Male*		70.5%	*CPB-time*	164 [147;183]	*Bleeding-24h*	370 [250;610]
*Date*		956 [649;1203]	*Clamp-time*	91 [81;106]	*Transfus*	24.6%
*COVID*		38.3%	*StS-time*	260 [230;290]	*N-transf*	0 [0;0]
*Age*		59.1 ± 13.3	*TOR-time*	370 [345;410]	*Hb-post*	11.3 ± 1.5
*Height*		173.0 ± 9.2	*OR-extub*	65.1%	*ICU-hours*	40 [20;45]
*Weight*		72.7 ± 13.5			*Post-days*	7 [6;9]
*BSA*		1.9 ± 0.2			*Hosp-days*	10 [9;12]
*BMI*		24.1 ± 3.3			*Home*	89.3%
*Hb-pre*		14.3 [13.6;15.0]				
*Rhythm*		14.8%				
*HTA*		45.6%				
*Diab-NID*		3.4%				
*Diab-ID*		0.7%				
*Resp-ins*		0.0%				
*Smoke*		22.8%				
*Endoc*		0.0%				
*ASA*	*I*	2.0%				
	*II*	28.2%				
	*III*	64.4%				
	*IV*	5.4%				
*EuroscoreII*		0.9 [0.7;1.2]				
*NHYA*	*I*	2.0%				
	*II*	71.1%				
	*III*	26.2%				
	*IV*	0.7%				
*EF-pre*		65 [60;68]				
*Mitral-reg*	*I*	0.7%				
	*II*	5.4%				
	*III*	94.0%				
*Mitral-st*		2.7%				
*Pulm-hypert*		45.0%				
*AoR*		18.1%				
*Creatinine*		0.93 [0.81;1.07]				
*Dialysis*		0.0%				

**Table 2 medicina-59-01568-t002:** Coefficients of the covariates and pseudo-*R^2^* for the fitted models for intraoperative variables. Covariates that were not included by the AIC for the respective model are indicated by “—” or are not shown at all. The significance codes of the coefficients are *** for *p*-value ∈ [0, 0.001], ** for *p*-value ∈ (0.001, 0.01], * for *p*-value ∈ (0.01, 0.05], and for *p*-value ∈ (0.05, 0.1].

		*CPB-Time*		*Clamp-Time*		*StS-Time*		*TOR-Time*		*OR-Extub*	
*Intercept*		+8.20	***	+5.34	***	+7.60	***	+6.50	***	−2.53 × 10^1^	
*Male*		−8.17 × 10^−2^		−9.39 × 10^−2^		−7.20 × 10^−2^	.	−8.70 × 10^−2^	***	—	
*Date*		−7.29 × 10^−4^	***	−6.83 × 10^−4^	**	−8.97 × 10^−4^	***	−5.83 × 10^−4^	***	−1.17 × 10^−2^	*
*Date^2^*		+3.36 × 10^−7^	**	+3.45 × 10^−7^	**	+4.15 × 10^−7^	***	+2.62 × 10^−7^	***	+4.88 × 10^−6^	*
*COVID*		—		—		—		—		—	
*Age*		—		−1.91 × 10^−2^	.	−1.77 × 10^−3^		—		−5.97 × 10^−2^	*
*Age^2^*		—		+1.76 × 10^−4^	*	—		—		—	
*Weight*		+5.72 × 10^−2^	**	+3.52 × 10^−3^	.	+8.86 × 10^−2^	**	+6.57 × 10^−2^	**	−1.99	**
*BSA*		−2.72	*	—		−1.28		—		—	
*BSA^2^*		—		—		−8.21 × 10^−1^	.	−8.4 × 10^−1^	**	+2.73 × 10^1^	**
*BMI*		−7.99 × 10^−2^	*	—		−1.15 × 10^−1^	*	−8.47 × 10^−2^	*	+2.64	**
*Rhythm*		+8.00 × 10^−2^	.	—		+6.36 × 10^−2^	.	—		—	
*HTA*		—		−7.27 × 10^−2^	.	—		—		—	
*Diab-ID*		—		—		—		+4.01 × 10^−1^	***	—	
*ASA*	*= II*	+3.23 × 10^−2^		+1.47 × 10^−1^		+4.41 × 10^−2^		—		+1.96 × 10^1^	
	*= III*	+1.25 × 10^−1^		+2.22 × 10^−1^		+9.85 × 10^−2^		—		+2.09 × 10^1^	
	*= IV*	+2.42 × 10^−1^	.	+3.84 × 10^−1^	*	+1.58 × 10^−1^		—		+3.79 × 10^1^	
*EuroscoreII*		—		—		+1.30 × 10^−2^	.	—		—	
*EuroscoreII^2^*		—		—		—		—		+5.38 × 10^−1^	*
*Mitral-reg*	*= II*	—		−5.26 × 10^−1^	*	—		—		—	
	*= III*			−3.76 × 10^−1^		—		—		—	
*Mitral-st*		+2.29 × 10^−1^	*	—		+1.50 × 10^−1^	*	+1.03 × 10^−1^	.	—	
*Pulm-hypert*		—		—		—		−4.13 × 10^−2^	*	—	
*AoR*		—		—		—		—		+1.11	.
Pseudo-*R*^2^		0.314		0.229		0.509		0.545		0.351	

**Table 3 medicina-59-01568-t003:** Coefficients of the covariates and pseudo-*R^2^* for the fitted models of postoperative variables *Bleeding-24h*, *Transfus*, *N-transf* and *Hb-post*. The structure of the table is as in Table 2.

		*Bleeding-24h*		*Transfus*		*N-Transf*		*Hb-Post*	
*Intercept*		+1.41 × 10^1^	*	+1.39 × 10^2^		+1.47 × 10^1^	**	+2.97	***
*Male*		+4.67 × 10^−1^	**	—		+1.18	**	—	
*Date*		—		−7.96 × 10^−3^	.	—		—	
*Date^2^*		—		+6.86 × 10^−6^	*	—		+5.36 × 10^−8^	**
*COVID*		−3.02 × 10^−1^	**	—		−7.14 × 10^−1^	*	+3.24 × 10^−2^	
*Age*		+6.46 × 10^−2^	*	+1.21 × 10^−1^	**	—		—	
*Age^2^*		−6.47 × 10^−4^	*	—		—		—	
*Height*		−6.33 × 10^−2^		−7.12 × 10^−1^	*	—		−3.46 × 10^−2^	**
*Weight*		−1.01 × 10^−1^	.	+3.99	**	+1.18	*	−4.43 × 10^−2^	**
*BSA*		—		—		—		+4.39	**
*BSA^2^*		+2.14		−4.39 × 10^1^	*	−1.74 × 10^1^	*	—	
*BMI*		—		−6.82	**	−1.61	*	—	
*Hb-pre*		—		—		−1.84 × 10^−1^	*	+3.21 × 10^−2^	***
*Rhythm*		—		−2.18	*	—		—	
*HTA*		—		−1.29	.	−9.07 × 10^−1^	*	—	
*Diab-NID*		+4.32 × 10^−1^		—		—		—	
*Smoke*		+2.56 × 10^−1^	.	+1.29	.	+1.62	***	−5.10 × 10^−2^	*
*ASA*	*= II*	—		+1.23 × 10^1^		—		—	
	*= III*	—		+1.19 × 10^1^		—		—	
	*= IV*	—		+1.53 × 10^1^		—		—	
*EuroscoreII*		+5.08 × 10^−1^	***	—		+1.50	***	—	
*EuroscoreII^2^*		−2.06 × 10^−2^	**	—		−7.21 × 10^−2^	***	—	
*EF-pre*		—		—		+3.36 × 10^−2^		—	
*Mitral-reg*	*= II*	+3.32 × 10^−1^		—		—		—	
	*= III*	+8.81 × 10^−1^		—		—		—	
*Pulm-hypert*		−2.69 × 10^−1^	*	—		−6.67 × 10^−1^	*	—	
*Creatinine*		—		−1.77 × 10^1^	**	−8.29	**	—	
*Creatinine^2^*		—		+8.69	**	+3.90	**	—	
Pseudo-*R*^2^		0.299		0.523		0.499		0.372	

**Table 4 medicina-59-01568-t004:** Coefficients of the covariates and pseudo-*R^2^* for the fitted models of postoperative variables *ICU-hours*, *Post-days*, *Hosp-days* and *Home*. The structure of the table is as in Table 2.

		*ICU-Hours*		*Post-Days*		*Hosp-Days*		*Home*	
*Intercept*		+8.42	**	+1.92	***	+2.65	***	−1.10 × 10^1^	
*Male*		−2.14 × 10^−1^		—		—		—	
*Date*		−2.43 × 10^−3^	***	—		−4.01 × 10^−4^		—	
*Date^2^*		+1.29 × 10^−6^	***	+1.19 × 10^−7^	*	+2.53 × 10^−7^		—	
*COVID*		—		—		—		—	
*Age*		—		+9.08 × 10^−3^	***	+4.27 × 10^−3^	.	+4.13 × 10^−1^	.
*Age^2^*		—		—		—		−3.75 × 10^−3^	*
*BSA*		−5.10		—		—		—	
*BSA^2^*		1.22		—		—		—	
*BMI*		+3.40 × 10^−2^	.	—		—		—	
*Hb-pre*		—		—		—		+8.45 × 10^−1^	**
*HTA*		−1.32 × 10^−1^		—		—		−3.05	*
*Smoke*		+2.01 × 10^−1^	.	—		—		−1.53	.
*ASA*	*= II*	+4.14 × 10^−1^		—		—		—	
	*= III*	+6.20 × 10^−1^	.	—		—		—	
	*= IV*	+1.03	*	—		—		—	
*EuroscoreII*		+2.60 × 10^−1^	*	—		+9.12 × 10^−2^	.	+1.62	.
*EuroscoreII^2^*		−1.12 × 10^−2^	*	—		−3.88 × 10^−3^		−8.90 × 10^−2^	.
*NHYA*	*= II*	−6.20 × 10^−1^	.	−4.93 × 10^−1^	**	−5.28 × 10^−1^	***	—	
	*= III*	−3.26 × 10^−1^		−5.43 × 10^−1^	**	−5.38 × 10^−1^	***	—	
	*= IV*	−1.37	*	−7.31 × 10^−1^	.	−7.17 × 10^−1^	*	—	
*EF-pre*		—		—		—		8.99 × 10^−2^	
*Mitral-reg*	*= II*	+8.11 × 10^−1^		—		—		—	
	*= III*	+2.18 × 10^−1^		—		—		—	
*Mitral-st*				—		—		−3.79	*
*Pulm-hypert*		+2.56 × 10^−1^	*	—		—		—	
*Creatinine*		+8.50 × 10^−2^		—		—		−2.10 × 10^1^	.
*Creatinine^2^*		—		—		—		+7.24	
Pseudo-*R*^2^		0.391		0.216		0.199		0.435	

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
