# Peer review of "Learning Curve Analysis of Robotic-Assisted Mitral Valve Repair with COVID-19 Exogenous Factor: A Single Center Experience"

_medicina, 2023, doi:10.3390/medicina59091568_

Round 1

Reviewer 1 Report

The authors have presented data for 149 patients that underwent RAMVS, irrespective of etiology and type of surgery. After analyzing covariates, the authors concluded that the learning curve had a positive impact on the trend of postoperative parameters. However, COVID19 pandemic did not significantly influence postoperative outcomes and learning curve.

I have few major concerns about this manuscript.

The presentation of the results is unclear. The results in the tables should be presented in a more understandable way.

All coefficients of the covariates are listed, irrespective of their importance.  I would suggest to show the results in a more clear way and, if possible, use graphical presentation of the results.

Discussion should rely more on the results. In example, weight and BSA (proportional variables) showed completely opposite impact on outcomes (row 339-344). The discussion does not follow the results.

English language level is satisfactory. These is a type feller in row 84.

Reviewer 2 Report

First I would like to congratulate the authors for their article in this interesting field. The authors generate an interesting overview regarding MV repair with robotic assisted technique. I suggest to accept the manuscript after minor english editing and formatting of tables wich can be designed a bit more structured / easier to read.

english editing required
